# Impacts of Varying Blood Flow Restriction Cuff Size and Material on Arterial, Venous and Calf Muscle Pump-Mediated Blood Flow

Steven B. Machek [1,2,*], Dillon R. Harris [1], Jeffery L. Heileson [1,3], Dylan T. Wilburn [1], Jeffrey S. Forsse [1,*] and Darryn S. Willoughby [1,4]

1 Integrated Laboratory of Exercise, Nutrition, and Renal Vascular Research, Department of Health, Human Performance, and Recreation, Robbins College of Health and Human Sciences, Baylor University, Waco, TX 76706, USA; dillon_harris1@baylor.edu (D.R.H.); jeffery.l.heileson.mil@army.mil (J.L.H.); dylan_wilburn1@baylor.edu (D.T.W.); dwilloughby@umhb.edu (D.S.W.)
2 Kinesiology Department, College of Health Sciences and Human Services, California State University, Monterey Bay, Seaside, CA 93955, USA
3 Nutrition Services Division, Walter Reed National Military Medical Center, Bethesda, MD 20814, USA
4 School of Exercise and Sport Science, Mayborn College of Health Sciences, University of Mary Hardin-Baylor, Belton, TX 76513, USA
* Correspondence: smachek@csumb.edu (S.B.M.); jeff_forsse@baylor.edu (J.S.F.);
  Tel.: +1-831-582-5455 (S.B.M.); +1-254-710-3900 (J.S.F.)

**Abstract:** Blood flow restriction (BFR) may become ineffective or potentially dangerous without sufficient standardization. The purpose of this investigation was therefore to (1) assess the viability of multiple sizes of a novel BFR cuff to determine arterial occlusion pressure (AOP) and (2) compare resting arterial, venous and calf muscle pump (cMP)-mediated blood flow between the aforementioned conditions and a commonly employed wide-rigid, tourniquet-style cuff. In randomized, counter-balanced, and crossover fashion, 20 apparently healthy males (18–40 years) donned a widely employed wide-rigid (WR) cuff, along with the largest (NE) and manufacturer-recommended sizes (NER) of a novel narrow-elastic cuff. Participants subsequently assessed AOP, as well as (at 80%AOP) arterial, venous, and venous cMP flow relative to baseline values via ultrasound. All analyses were performed at a significance level of $p < 0.05$. Analyses revealed a significant condition effect for AOP ($p < 0.001$; $\eta_P^2 = 0.907$) whereby WR was significantly lower than both NE and NER; in addition, the latter two did not differ. Compared with baseline, there were no statistically significant differences between cuffs for either arterial or cMP-mediated blood flow. Unsurprisingly, no participants demonstrated venous blood flow at 80% AOP. These findings support the viability of a novel narrow-elastic BFR product, evidenced by consistent AOP acquisition and equivocal blood flow parameters.

**Keywords:** arterial occlusion pressure; limb occlusion pressure; calf muscle pump; ultrasound; arterial flow; subjective discomfort

## 1. Introduction

Blood flow restriction (BFR) training is commonly employed to elicit skeletal muscle hypertrophy and commensurate strength gains in a multiplicity of demographics, ranging from healthy athletic populations to rehabilitation patients and the elderly [1,2]. Specifically, this training modality functions via full venous and partial arterial occlusion of the proximal limb(s) to generate a localized hypoxic environment that preferentially activates high threshold motor units [3,4]. Nevertheless, this method is not without potential risks, possibly augmenting sympathetic output to elevate heart rate, blood pressure, and cardiac output (CO), but also increasing arterial resistance and reducing venous capacitance [5–7]. Individualized pressure prescriptions are therefore imperative, as excessive restriction may lead to a myriad of complications in at-risk populations (i.e., chronic hypertension and

peripheral artery disease), including significant ischemia, deep vein thrombosis secondary to unrestrained venous compression, and other deleterious cardiovascular events [7].

While previous investigations have determined that elastic knee wraps can be employed to practically and economically execute BFR training, this method is difficult to standardize, and it is often subject to the participants' perceived discomfort—and is thus less reliable—when determining sufficient occlusion [1,2,8,9]. Furthermore, attempts to standardize cuff pressures to brachial systolic blood pressure are also inconsistent and especially so when utilizing a cuff other than the one desired to perform BFR training [10]. The most sensible approach may be the implementation of a relative arterial occlusion pressure (AOP). This value is determined by temporarily inflating the cuff to the point of complete occlusion (100% AOP) and subsequently reducing compression to a percentage (typically 40–80%) of AOP during exercise [10,11]. Multiple factors determine individual AOP, including limb circumference as well as cuff material and size [1,2,4,10,12,13]. Cuffs characterized by smaller widths, elastic materials, and application on larger limbs require greater absolute pressures to attain occlusion [4,12–14].

A novel, commercially available BFR cuff characterized by a narrow-elastic design (similar to the empirically substantiated Kaatsu devices) and multiple pneumatic chambers is currently supported by a small number of investigations [5,6]. Previous authors stipulate that this product ostensibly exerts favorable hemodynamics relative to often utilized and so-called wide-rigid, tourniquet-style cuffs [5,6]. Incidentally, Stray-Gundersen et al. [6] corroborate this notion, demonstrating that the novel cuff produced lower heart rate and blood pressure responses relative to a wide-rigid product during a walking interval series; however, these hemodynamic parameters and blood lactate concentrations were surprisingly not statistically different from a non-cuffed control. Although these findings may have been a result of the implementation of arbitrary cuff pressures between conditions and the omission of a standardized AOP, the authors contend that these "favorable" results are largely promoted via the proprietary cuff design [6]. Specifically, the narrow-elastic material supposedly attenuates mechanically mediated vascular resistance by minimizing working muscle compression, thus allowing expansion, peripheral vasodilation, and intermittent muscle pump-mediated venous return [5,6].

The claimed benefits of this novel, multi-chambered narrow-elastic cuff design and its potential implications towards a plethora of healthy and clinical demographics warrant further investigation prior to more widespread clinical and recreational implementation. It is hitherto unknown whether AOP can be determined with this cuff to both ensure standardization and general venous blood flow cessation, especially considering previous evidence whereby arterial occlusion could not be reached with similar narrow-elastic devices [13]. Furthermore, the calf muscle pump (cMP) represents one of the most physiologically imperative contributors towards dynamic exercise blood flow maintenance, and its role is wholly undescribed in BFR training [15,16]. The purpose of this investigation was therefore to (1) determine the viability of this novel BFR cuff design's recommended and largest sizes on AOP assessment and (2) compare the impact of the aforementioned sizes and a widely employed wide-rigid, tourniquet-style cuff on cMP-mediated blood flow. We hypothesized that AOP would only be attained amongst all participants in the wide-rigid and largest narrow-elastic cuffs. Lastly, we hypothesized that the wide-rigid cuff would prevent all blood flow via the cMP, without discernable effects from either narrow-elastic condition in this regard.

## 2. Materials and Methods

### 2.1. Experimental Approach to the Problem

In this randomized, crossover (within-subject design) investigation, participants visited the laboratory on a single occasion for visits lasting ~90 min. After initial screening, participants laid supine, whereby baseline arterial, venous, and (venous) calf toe–tip movement-derived muscle pump blood flow assessments were sequentially conducted via ultrasound. Participants were subsequently randomized in crossover and counter-

balanced fashion to a nylon-based, wide-rigid (WR) cuff, as well as the largest (NE) and manufacturer-recommended (NER) sizes of a novel, multi-chambered, narrow-elastic BFR cuff product. Successful inflation of each cuff condition on the participants' right leg was followed by arterial occlusion pressure (AOP) determination and each of the aforementioned ultrasound-mediated blood flow measurements in the previously listed sequential order. A 10 min supine rest period separated both the baseline from the cuffed measurements and also each experimental cuff condition [12]. A general overview of the present investigation is visualized in Figure 1.

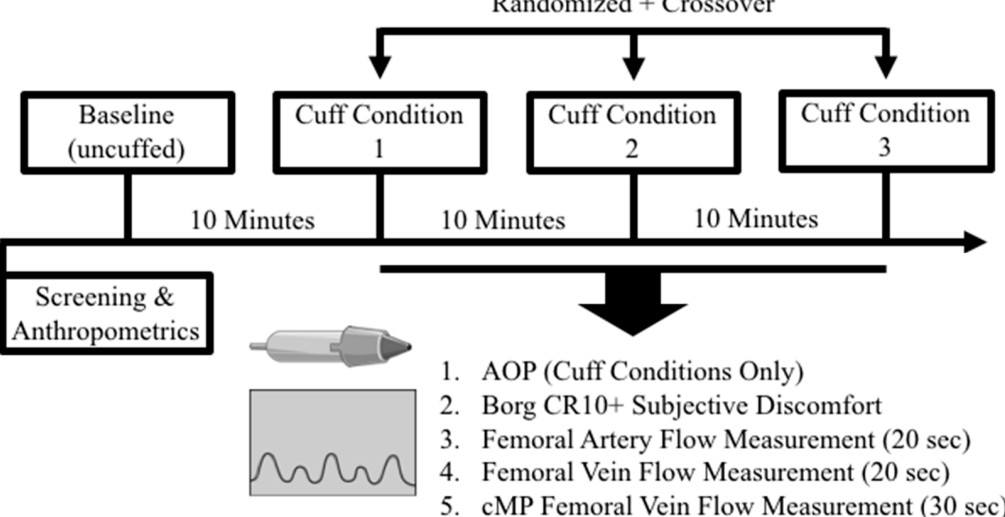

**Figure 1.** General overview of study methodology and timeline. AOP = arterial occlusion pressure; cMP = calf muscle pump.

### 2.2. Participants

Twenty apparently healthy and resistance- trained males between the ages of 18 and 40 volunteered from the central Texas area and completed this study. Females were not included in the present investigation due to inherent sex-specific differences in vessel morphology and/or hemodynamic responses relative to males [17,18]. Additionally, the present study somewhat acted as a pilot trial to a larger investigation aimed at determining possible BFR and betaine supplement interactions; it was therefore impractical to recruit females due to their sex-specific betaine metabolism differences [19–21]. Total body mass (kg) and height (cm) were determined on a standard dual-beam balance scale (Detecto Bridgeview, IL, USA) and all participant descriptive data that were recorded following successful screening are depicted in Table 1. Specifically, all participants were required to actively participate in resistance training as per the definition of the American College of Sports Medicine (ACSM) ($\geq$30 min exercise, $\geq$3 days per week, over the last 3 months) and a minimum of 1 day per week of leg-focused training all over the year prior to the onset of the investigation [22]. Additionally, all participants were unaccustomed to BFR training and specifically had no experience training with any of the cuff products employed in the present investigation. The use of blood thinning (e.g., Warfarin, Jantoven, etc.), heart, pulmonary, thyroid, antihypertensive, anti-hyperlipidemic, hypoglycemic, endocrinologic (e.g., prednisone, Ritalin, Adderall, etc.), or neuromuscular/neurological medications were further prohibited for eligibility. Furthermore, all participants were required to be non-smokers, as well as having a resting (systolic and diastolic) blood pressure of <140/90 mmHg and a resting heart rate of <90 bpm following 10 min of seated rest in a quiet room [12]. The aforementioned resting hemodynamic variables were assessed using an automated non-invasive blood pressure unit (E-sphyg™ II, American Diagnostic Corporation, Hauppauge, NY, USA). All eligible participants signed university-approved written informed consent documents, and approval was granted by the Institutional Review Board at Baylor University, reference #1676709, approval date: 1/26/2021. In addition, all

experimental procedures involved in the study conformed to ethical considerations of the Helsinki Code.

**Table 1.** Participant anthropometric and resting hemodynamic characteristics.

| Subject Characteristics (n = 20) | Mean $\pm$ SD |
|---|---|
| Age (years) | $25 \pm 5$ |
| Height (cm) | $175.2 \pm 6.0$ |
| Weight (kg) | $85.2 \pm 12.0$ |
| Right Leg Circumference (cm) | $61.3 \pm 6.3$ |
| Resting SBP (mmHg) | $125 \pm 13$ |
| Resting DBP (mmHg) | $75 \pm 13$ |
| Resting Heart Rate (bpm) | $66 \pm 11$ |

Bpm = beats per minute; DBP = diastolic blood pressure; SBP = systolic blood pressure.

### 2.3. Cuff Procedures

Before visiting the laboratory, participants were instructed to both abstain from strenuous exercise and fast (including caffeine consumption) for at least 24 and 6 h, respectively [23]. Following successful screening and anthropometric measurements, all participants rested 10 min in a supine position inside a quiet, temperature-controlled room (22.2–23.3 °C) before baseline arterial, venous, as well as venous MP blood flow parameters were sequentially assessed [6,12]. Subsequently, the participants were then randomized into the following three groups based on conditions in counter-balanced and crossover fashion: a widely implemented, nylon-based, wide-rigid (WR; Hokanson, Bellevue, WA, USA) cuff with a single pneumatic chamber, the largest (NE) cuff, and the recommended (NER) cuff size of a novel, narrow-elastic, multi-pneumatic chambered cuff (B3 Sciences Inc., Frisco, TX, USA) (see Figure 2 and Table 2 for individual product visualization and dimensions, respectively) [5,6,24,25]. In brief, the NE and NER cuffs are derived from the same manufacturer, whereby the former was fixed to the largest size (Size #4) and the latter was either Size #3 or #4. An additional 10 min of quiet, un-cuffed supine rest separated each condition as supported by the previous literature [12]. All cuffs were placed and tightened immediately distal to the inguinal crease on the participant's right leg (marked with a pen for successive cuff condition placement) for subsequent AOP assessment, as well as determination of the aforementioned blood flow parameters [26].

### 2.4. Ultrasound Arterial Occlusion Pressure and Blood Flow Assessments

A small amount of gel was placed on the inner thigh of each participant's right leg and both the superficial femoral artery and the superficial femoral vein were sequentially located via two-dimensional ultrasound (B Mode Logic S7 Expert/Pro Ultrasound™, General Electric Systems®) and 9L-D probe at a 7 MHz frequency in a transverse and sagittal view. In each case, assessments strictly adhered to the following order: superficial femoral artery detection and blood flow measurement, superficial femoral vein detection and blood flow measurement, and femoral vein cMP blood flow measurement. Additionally, anatomical landmarks were used and marked 10 cm above the medial epicondyle of the femur with pen to optimize clarity and standardization [27]. All cuff types were inflated at 10 mmHg increments (rapid cuff inflator (E 20™, Hokanson®, Hokanson Inc., Bellevue, WA, USA) for the WR condition and via manual hand pump for NE and NER conditions) until femoral artery blood flow could no longer be discerned by pulse wave velocity and by the absence of auditory auscultatory signals (i.e., AOP). Baseline arterial and venous mean and peak blood flow were digitally recorded for 20 s via Doppler radar video with signals corrected at an insonation angle of 60° and a steer angle of 15° adjusted to the axis of blood flow. Following the occlusion verification (1–2 min) of each experimental condition, cuff pressure was immediately released to 80% AOP and all aforementioned blood flow parameters were similarly assessed separately and in the aforementioned sequence. Briefly, 80% AOP was selected as an empirically supported partial arterial occlusion pressure to more reliably elicit the metabolic stress necessary to glean positive BFR exercise adaptations [28,29].

NER cuff pressure following successful AOP determination was notably specific to the manufacturer's recommendations (250 mmHg and 300 mmHg for Size #3 and #4 cuffs, respectively), and thus did not employ a relative AOP. To assess baseline and cuff condition-specific cMP-mediated femoral vein (MP) blood flow parameters, the participants were instructed to perform non-resisted, supine laying "tip-toe" plantar flexion movements to a metronome-facilitated 2-1-2-1 (concentric top range of motion-eccentric bottom range of motion) cadence, and Doppler video was recorded for 30 s following venous blood flow at baseline and every condition [30]. The total duration at baseline and 80% AOP to assess all blood flow indices was ~90–120 s. Subsequently, arterial, venous, and MP mean and peak blood flow were analyzed offline using the Vascular Research Tools$^{TM}$ software (Medical Imaging Applications, Coralville, IA, USA), and images were captured at a rate of 6.0 frames per second [26]. The aforementioned cuff application, ultrasound procedures, and flow analyses were repeated for all subsequent conditions, as well as performed by the same technician across every participant to reduce inter-rater variability.

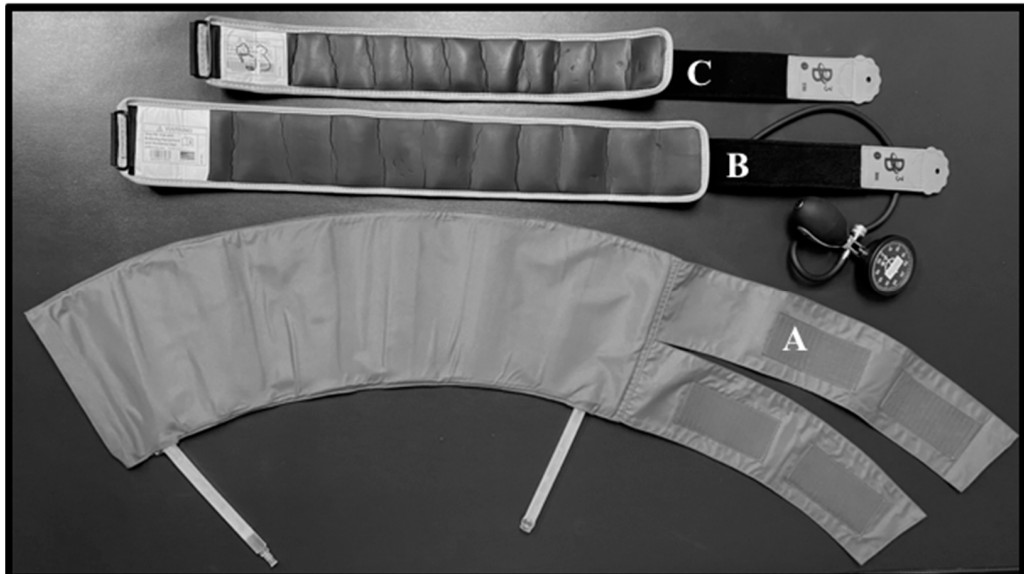

**Figure 2.** Visualization of wide-rigid (WR; A), as well as Size #4 (largest; NE; B) and #3 (C) narrow-elastic cuffs conditions. The cuff employed for the NER condition was predicated on circumference measurements taken at the site of application (i.e., on the participant's right leg and immediately distal to the inguinal ligament). NE = narrow-elastic; NER = narrow-elastic recommended; WR = wide-rigid.

**Table 2.** Cuff condition product specifications.

| Code | Product | Length (cm) | Width (cm) | Max Pressure (mmHg) |
|------|---------|-------------|------------|---------------------|
| WR | Hokansen CC17 | 108.0 | 18.0 | 300 |
| NE | B3 Size #4 | 60.96 | 9.53 | 500 |
| NER * | B3 Size #4 | " | " | " |
| | B3 Size #3 | 50.80 | 7.62 | 500 |

Max = maximum; NE = narrow-elastic; NER = narrow-elastic recommended; WR = wide-rigid.* Recommended sizing for B3 Size #3 and #4 cuffs are leg circumferences of 44.5–60 cm and 60–78.5 cm, respectively.

## 2.5. Subjective Discomfort

Additionally, perceptual discomfort between conditions was assessed via the Borg CR10+ scale. Baseline values were obtained immediately following the initial 10 min supine resting period, whereby the scale was explained in detail akin to the methods described by Loenneke et al. [14] and Buckner et al. [12]. Briefly, the Borg CR10+ scale assesses subjective

discomfort on a scale from 1 (no discomfort) to 10 (maximal discomfort); the latter rating is also anchored by the participant's greatest memory of discomfort and therefore can be exceeded if the present methods exceed that experience [14]. Participants were asked to describe their perceived discomfort for each condition immediately preceding blood flow parameter assessment. It is worth to mention that although the Borg CR10+ scale is not specific to tightness, it is commonly employed within the BFR literature, and the previously reported tightness-specific scale unreliably estimates blood flow restriction [12,14,31].

### 2.6. Statistical Analyses

Previous a priori power analysis using G*Power 3.1.9.6 (Kiel University, Kiel, Germany) determined that a total of 18 participants was necessary to achieve an anticipated $\eta^2 = 0.385$ and power $(1-\text{ß}) = 0.80$ at $\alpha = 0.05$. All variables were tested for normality and homogeneity of variance using the Shapiro–Wilk test and Mauchly's test of Sphericity, respectively, before continuing the subsequent analysis. AOP and blood flow parameter percent changes (mean and peak arterial ($\%A_{MEAN}$; $\%A_{PEAK}$), venous ($\%V_{MEAN}$; $\%V_{PEAK}$), and cMP-mediated ($\%MP_{MEAN}$; $\%MP_{PEAK}$) blood flow) relative to baseline were assessed via factorial one-way analysis of variance (ANOVA) with repeated measures for condition (WR, NE, NER). Notably, relative changes in blood blow were employed to simplify subsequent interpretations and have been previously employed in the prior literature [32–34]. If significant main condition effect was present, pairwise comparison analyses were used with a Bonferroni adjustment for alpha inflation. Furthermore, nominal subjective discomfort data (Borg CR10+ scores for baseline and WR, NE, and NER) as well as all prior dependent variables failing to meet normality and/or homogeneity assumptions were assessed via nonparametric Friedman's ANOVA with follow-up Wilcoxon signed rank tests if a significant effect was observed. Eta squared ($\eta^2$) was employed to estimate the proportion of variance in the dependent variables explained by the independent variable. Eta squared effect sizes were determined to be weak = 0.17, medium = 0.24, strong = 0.51, very strong = 0.70 [35]. Additionally, Kendall's W coefficient of concordance was used as an estimate of agreement for nonparametric data, whereby 0–0.20 = slight agreement, 0.20–0.39 = fair agreement, 0.40–0.59 = moderate agreement, 0.60–0.79 = substantial agreement, and >0.80 = almost perfect agreement [36]. All analyses were performed in SPSS V.27 (IBM Corporation; Armonk, NY, USA) at an alpha of $p < 0.05$ and values are reported as means $\pm$ standard deviations (SD). Lastly, confidence intervals (CI) for significant comparisons are reported as 95% CI (lower bound, upper bound).

### 3. Results

#### 3.1. Arterial Occlusion Pressure

Hemodynamic results are displayed in Table 3. NE and NER sizing was equal for 65% of participants (measured within the lower-limb circumference range for the largest (Size #4) cuff). Furthermore, 20% of participants had NER recommended pressures that were $\geq 100\%$ measured AOP (see Table 3). All four participants in this regard were assigned to the largest cuff (Size #4) for the NER condition. The assumption of sphericity was violated for AOP and thus the Greenhouse–Geisser correction was employed. Consequently, one-way factorial analyses revealed a very strong significant main effect for cuff type on AOP ($p < 0.001$; $\eta^2 = 0.907$), whereby WR was significantly lower than both NE ($p < 0.001$; CI [$-264.812$, $-184.488$]) and NER ($p < 0.001$; [$-263.013$, $-183.787$]), of which the latter two did not differ ($p = 1.000$) (see Figure 3). The intra-measurement coefficient of variation (CV%) values for WR, NE, and NER AOP assessment were 10.9%, 17.6%, and 17.5%, respectively. Consequently, the average condition-collapsed CV% was $15.3 \pm 3.9\%$.

**Table 3.** (Top) Condition-associated arterial, venous, and venous MP blood flow parameters, including baseline, WR, NE, and NER mean and peak blood flow expressed in m•s$^{-1}$.

| Arterial & Venous Blood Flow Parameters | | | | | | |
|---|---|---|---|---|---|
| Mean ± SD m•s$^{-1}$ | Mean A Flow | Peak A Flow | Mean V Flow | Peak V Flow | Mean MP Flow | Peak MP Flow |
| Baseline | 0.101 ± 0.065 | 0.328 ± 0.101 | 0.103 ± 0.065 | 0.156 ± 0.086 | 0.215 ± 0.097 | 0.325 ± 0.133 |
| WR | 0.064 ± 0.062 | 0.283 ± 0.152 | 0.000 | 0.000 | 0.087 ± 0.049 | 0.190 ± 0.067 |
| NE | 0.051 ± 0.028 | 0.238 ± 0.050 | 0.000 | 0.000 | 0.066 ± 0.044 | 0.177 ± 0.106 |
| NER | 0.041 ± 0.019 | 0.215 ± 0.056 | 0.000 | 0.000 | 0.093 ± 0.082 | 0.200 ± 0.126 |
| NER Exceeding AOP * | | | | | | |
| Participant Code | NER RecP (mmHg) | NER AOP (mmHg) | NER RecP % AOP | | | |
| P4 | 300 | 250 | 120 | | | |
| P5 | 300 | 280 | 107 | | | |
| P7 | 300 | 300 | 100 | | | |
| P17 | 300 | 300 | 100 | | | |

A = arterial; AOP = arterial occlusion pressure; MP = venous muscle pump; NE = narrow-elastic; NER = narrow-elastic recommended; RecP = recommended pressure; V = venous; WR = wide-rigid. *: (Bottom) Four participants' NER (manufacturer) recommended pressure either met or surpassed the NER-assessed AOP, therein indicating full arterial occlusion of the femoral artery.

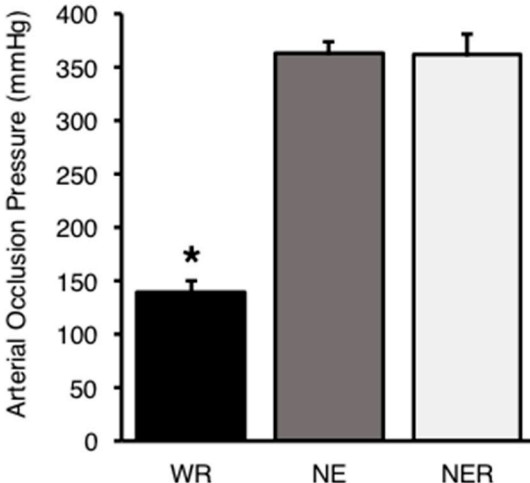

**Figure 3.** Comparison of arterial occlusion pressures (AOP; mmHg) amongst cuff conditions. * Significant main condition effect, whereby NE (363.4 ± 64.0 mmHg) and NER (362.2 ± 63.5 mmHg) displayed significantly higher AOP values relative to WR (138.8 ± 15.1 mmHg), without any discernable differences between each other. A = arterial; AOP = arterial occlusion pressure; MP = venous muscle pump; NE = narrow-elastic; NER = narrow-elastic recommended; WR = wide-rigid.

### 3.2. Blood Flow Assessments

Examples of ultrasound arterial and cMP pulse waves amongst all conditions can be viewed in Figure 4. A total of 5% of participants did not have arterial blood flow at 80% AOP in the NE condition. Likewise, 40% (8/20), 30% (6/20), and 40% (8/20) of participants displayed no discernable calf muscle pump-mediated blood flow at 80% AOP in the WR, NE, and NER conditions, respectively. Venous blood flow was unable to be detected amongst any condition at 80% AOP. Furthermore, the analyses demonstrated no significant differences between conditions for mean percent changes from baseline (resting) blood flow (m•s$^{-1}$) for arterial ($p$ = 0.291; WR = 88.9 ± 81.4%, NE = 78.7 ± 58.4%, and

NER = 46.9 ± 21.4%) or venous MP ($p$ = 0.565; WR = 52.9 ± 49.1%, NE = 40.2 ± 52.1%, and NER = 48.9 ± 53.1%). Similarly, there was no significant effect for cuff condition on peak arterial ($p$ = 0.565; WR = 100.6 ± 80.1%, NE = 81.1 ± 33.9%, and NER = 71.3 ± 17.7%) or venous MP values relative to baseline ($p$ = 0.368; WR = 64.7 ± 32.2%, NE = 54.3 ± 30.6%, and NER = 65.2 ± 46.6%).

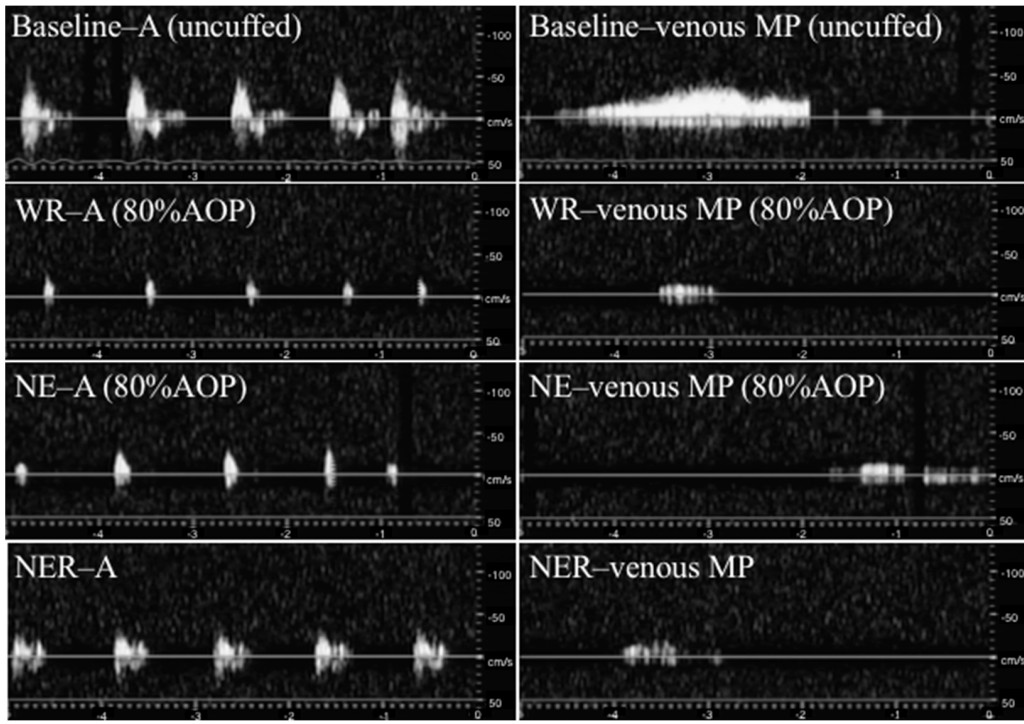

**Figure 4.** Visualization sample of assessed arterial and venous MP ultrasound pulse waves for baseline and all cuff conditions at 80% AOP (WR and NE) and at manufacturer-recommended cuff pressure (NER; 250 mmHg for displayed participant). It should be noted that no cuffed conditions demonstrated discernable blood flow, and thus neither these measurements nor baseline venous blood flow are depicted above. A = arterial blood flow; MP = venous muscle pump; NE = narrow-elastic; NER = narrow-elastic recommended; WR = wide-rigid.

*3.3. Subjective Discomfort*

Nonparametric analyses revealed a significant effect for subjective discomfort ($p$ < 0.001; Kendall's W = 0.575). Subsequent Wilcoxon Signed Ranked Tests demonstrated that all cuff conditions displayed greater CR10+ versus baseline without significant differences between cuffs (WR-NE ($p$ = 0.129), WR-NER ($p$ = 0.142), NE-NER ($p$ = 0.565)).

## 4. Discussion

The present investigation is the first to assess the viability of a newly designed narrow-elastic BFR product and whether it both reliably induces full venous and temporary full arterial occlusion for cuff pressure standardization purposes. Additionally, we are unaware of any prior literature that attempts to discern the impact of cMP-mediated venous blood flow amongst this or any commonly employed wide-rigid/tourniquet-style cuffs. Contrary to our initial hypothesis, all cuff conditions were able to fully occlude superficial femoral artery (and venous) flow amongst every participant in the current study, albeit both narrow-elastic cuffs (NE and NER) did so with significantly higher pressures compared to the WR condition. These findings directly contradict those of Weatherholt et al. [13], whereby they were unable to sufficiently determine lower limb AOP using the commonly employed narrow-elastic Kaatsu products relative to a tourniquet-style wide-rigid cuff. Moreover, our data somewhat oppose the notion that wider cuffs require less pressure to elicit arterial occlusion, evidenced by nearly identical means and variance between the two narrow-elastic

conditions. Although this equivocal finding is likely due to the high percentage (65%) of participants sharing the largest cuff size as both NE and NER conditions, it is noteworthy to remark that this ostensible cuff width-and-AOP interaction is purely supported by cuff material (wide-rigid versus narrow-elastic) or (wide-rigid product only) cuff size comparisons [12,14,37]. We therein posit the necessity for future research to elucidate the specific impacts of varying narrow-elastic cuff widths. To the authors' knowledge, the current study is interestingly the first to reliably and consistently assess narrow-elastic cuff AOP; the preponderance of preceding studies utilizing similarly designed Kaatsu bands have operated with a single standardized (i.e., arbitrary) pressure or were unable to dependably determine arterial occlusion amongst all participants [12,13,19,21,38–41]. Moreover, employing arbitrary pressures may result in excessive arterial blood flow restriction and subsequently deleterious cardiovascular complications [5–7,42]. While these aforementioned investigations rarely exceed prescribed pressures of 300 mmHg, the current study illustrates that four participants reached or exceeded their AOP threshold (as high as 120% AOP) using the cuff manufacturer's arbitrary recommended pressure of 300 mmHg [38–41]. The potentially concerning implications of this discovery underscore the importance of standardized procedures amidst BFR-induced pressure application protocols in not only facilitating adequate training stimuli by attenuating premature fatigue, but also preventing adverse cardiovascular events amidst predisposed user demographics.

Notwithstanding that all experimental conditions were able to completely occlude resting venous blood flow, the current investigation is the first to compare varying cuff type (sizes and materials) arterial blood flows [15]. Consequently, we demonstrated that two sizes of a novel, narrow-elastic BFR product did not significantly differ with regard to mean and peak arterial flow relative to a consistently employed, empirically supported wide-rigid Hokanson cuff. This evidence is reassuring, substantiating that the present novel, narrow-elastic products induce the partial arterial and full venous occlusion that fosters BFR-mediated adaptation [1]. Conversely, any potential arterial blood flow differences may have been masked by the large within-subject variability observed between cuff conditions. These discrepancies may nonetheless be largely explained by the impacts of individual anthropometrics, as well as differing degrees of cuff size-associated limb—and thus vessel—compression [2,10,13,43,44]. Along with the equivocal blood flow parameters, subjective discomfort using the Borg CR10+ scale incidentally did not significantly differ between the conditions. Loenneke et al. [14] presents similar outcomes, employing the same discomfort scale to compare equivalent-width elastic or nylon cuffs (5 cm) amidst knee extensor exercise. Furthermore, Buckner et al. [12] found no CR10+ discomfort differences at rest but saw greater values favoring the thinner 3 cm wide elastic condition in the subsequent sets (all but the first of four sets) compared with a wider 5 cm nylon cuff. Although the authors in the aforementioned trial ostensibly credit the increased subjective scores to the greater relative (%AOP) pressures required with narrow-elastic cuffs, a potential and additional reason we failed to reach a statistical significance between any condition may have been our omission of an exercise intervention [12].

The cMP is the most physiologically imperative existing muscle pump due to a relatively large capacitance and concomitantly high pressures generated; it is ultimately facilitated by muscle contraction-mediated blood vessel compression (particularly veins), and subsequent muscle relaxation suctions and/or tethers the blood vessels open to augment venous return [16,45]. Prior to the current investigation, the potential implications of the cMP have been wholly undetailed and have only been hypothetically considered amidst the existing BFR literature [15]. Once again contradicting our hypothesis, these data substantiate that toe-tip movement-mediated venous MP flow occurs without significant between-cuff differences. Although several participants incidentally failed to demonstrate any detectable venous MP blood flow, this absence was consistent between all cuff types (6/20 participants in NE and 8/20 participants for both WR and NER). Stray-Gundersen et al. [6] and Wooten et al. [5] previously claimed that the proprietary design of the currently investigated novel, multi-pneumatic chambered, narrow-elastic

product uniquely facilitates an attenuated arterial resistance and peripheral vasodilation to ultimately foster muscle-pump-mediated venous return. Conversely, we provide evidence that venous MP blood flow is potentially a common phenomenon, as it occurred with both the narrow-elastic and wide-rigid cuffs. Previous authors have mixed interpretations as to whether the cMP plays a vital or inconsequential role in the maintenance of muscular performance [15,16,46]. Kilgas et al. [15] and Singer et al. [47] cumulatively present data suggesting that higher cuff pressure-mediated attenuations in overall limb blood flow may be rescued by concomitant enhancements in cMP-associated venous return. Therefore, we posit that this phenomenon possibly permits the muscular endurance necessary to achieve the higher exercise volume commonly characterized by BFR exercise and its subsequently associated beneficial adaptations.

The primary limitation of the current investigation is the absence of both an exercise intervention and central hemodynamic parameters alongside each cuff condition. Moreover, although AOP determination and blood flow parameters are most easily, accurately, and reliably measured in a resting supine position, our ability to extrapolate these data to an exercising model is limited [37,48]. We can nevertheless likely infer that the hemodynamic factors including arterial blood flow would become augmented due to enhancements in sympathetic outflow along with concomitant mean arterial pressure and CO elevations [7,49]. Venous MP flow may also see robust comparative alterations during an exercise model; this is especially so considering physical activity is commonly completed in an upright position, potentiating gravitational force-mediated lower limb blood pooling and associated enhancements in cMP-derived venous return [45]. Incidentally, exercise-derived blood flow augmentations can impact AOP accuracy, ultimately decreasing the relative pressure exerted by the cuff [49,50]. This consideration may be wholly arbitrary, however, as these formerly mentioned pressure attenuations are small and thus unlikely to bring the total relative prescription below the currently supported working (40–80% AOP) "effective range" [10,51].

Although our data suggest the presently investigated novel product may be an effective option to induce BFR-associated muscular adaptations, no evidence hitherto substantiates their use in upper body interventions. Nevertheless, the current cuffs and the extensively employed Kaatsu bands share a similar narrow-elastic design [5,6]. One can therefore reasonably assume somewhat equivalent outcomes given the latter's preponderance of the literature substantiating both positive upper and lower body parameters [40,52–55]. Regardless, the proposed utility of the present narrow-elastic product is wholly constrained beyond the resting conditions of the present trial and the two aforementioned investigations led by Stray-Gundersen et al. [6] and Wooten et al. [5] that employed walking and yoga exercise, respectively. Future research is therefore tasked with examining this novel cuff design's efficacy with regard to both upper body outcomes and ideally amidst a resistance-training intervention.

## 5. Conclusions

Overall, the present findings further substantiate the notion that various BFR product types have the potential to equally elicit positive adaptation [10]. Our data specifically reinforce that BFR protocol standardization—optimally via AOP assessment—is imperative when applying cuff pressure and that arbitrarily prescribed pressures may produce deleterious consequences [7]. Furthermore, the current investigation uniquely underscores a putative role regarding the cMP in BFR exercise and its potential to facilitate sufficient training volume [16]. As the associated research and industry prodigiously grow, it is abundantly clear that research has a fundamental responsibility to continually ensure both efficacy and safety amongst existing and emerging BFR-implementing strategies.

**Author Contributions:** Conceptualization, S.B.M. and J.S.F.; Methodology, S.B.M., J.S.F. and D.S.W.; Formal analysis, S.B.M. and J.S.F.; Investigation, S.B.M., D.R.H., J.L.H., D.T.W. and J.S.F.; Resources, S.B.M. and J.S.F.; Data curation, S.B.M. and J.S.F.; Writing—original draft preparation, S.B.M. and J.S.F.; Writing—review and editing, S.B.M., D.R.H., J.L.H., D.T.W., J.S.F. and D.S.W.; Project administration, S.B.M., J.S.F. and D.S.W.; Funding acquisition, S.B.M. and D.S.W. All authors have read and agreed to the published version of the manuscript.

**Funding:** The present research was partially funded by the American College of Sports Medicine—Texas Chapter, Student Research and Development Award, as well as the Baylor University Health, Human Performance, and Recreation Doctoral Dissertation Research Grant.

**Institutional Review Board Statement:** This study was conducted in accordance with the Declaration of Helsinki and approved by the Institutional Review Board of Baylor University (protocol code #1676709, approval date 1/26/2021).

**Informed Consent Statement:** Informed consent was obtained from all subjects involved in the study.

**Data Availability Statement:** The data presented in this study are available on request from the corresponding author.

**Acknowledgments:** The authors would like to express the upmost gratitude to Joel and Sandy Delaney from Underground Performance in Waco, TX, for their continuous support. Through their sustained and genuine interests in our research, we are guaranteed a perpetually exceptional audience.

**Conflicts of Interest:** S.B.M. is registered as a B3 BFR Certified Coach under B3 Sciences (Frisco, TX, USA). The results of the present study are presented clearly, honestly, and without fabrication, falsification, or inappropriate data manipulation. Moreover, these results do not constitute endorsement by the ACSM.

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
