# Peer review of "Impacts of Varying Blood Flow Restriction Cuff Size and Material on Arterial, Venous and Calf Muscle Pump-Mediated Blood Flow"

_oxygen, doi:10.3390/oxygen3020014_

Round 1

Reviewer 1 Report

The manuscript addresses the necessary topic of "Blood Flow Restriction". The manuscript is elaborated carefully in all its parts and brings the necessary results, beneficial for practice.

I only have a few minor recommendations:

Regarding the methodology: it is necessary to describe in more detail how the participants were selected, it would also be appropriate to indicate whether they had experience with BFR. I recommend adding a picture that will demonstrate the application of "cuffs" on the participant.

Explanations of abbreviations need to be added to all tables for greater clarity.

Limitations, which are listed together with the conclusion, I recommend reassigning to the discussion chapter.

I recommend keeping the conclusion as a separate chapter and writing this more concisely.

Minor editing required

Author Response

Dear Reviewer #1,

We are sincerely appreciative of both your positive comments and constructive feedback. We have made earnest attempts to implement edits where you have indicated whenever possible. Although we were not able to make changes based on every suggestion, we hope our reasons are satisfactory and invite you to address any additional feedback items you may have.

The manuscript addresses the necessary topic of "Blood Flow Restriction". The manuscript is elaborated carefully in all its parts and brings the necessary results, beneficial for practice.

I only have a few minor recommendations:

Regarding the methodology: it is necessary to describe in more detail how the participants were selected, it would also be appropriate to indicate whether they had experience with BFR. I recommend adding a picture that will demonstrate the application of "cuffs" on the participant.

Response: We have now added additional information to the methods section that indicates that our participant pool was taken from the central Texas area. Please let us know if this added detail is sufficient or there was a particular piece you were looking for and we would be happy to provide it. Furthermore, we have now included that our participants were unaccustomed to BFR as a whole and had no experience with the cuff products used in this investigation. Lastly, it would be unfortunately difficult to obtain an application figure at this given time because the first author (SBM) has moved institutions and thus no longer has access to all of the cuff conditions. We apologize for this inconvenience and hope that it does not majorly hinder your study interpretations.

Explanations of abbreviations need to be added to all tables for greater clarity.

Response: We apologize for any confusion but contend that all tables – and figures –  in the previous manuscript have legends to define their respective acronyms directly underneath them. Please do not hesitate to let us know if you cannot see the legend for any reason.

Limitations, which are listed together with the conclusion, I recommend reassigning to the discussion chapter.

Response: The limitations section has been moved to the discussion and the concluding paragraph retains its place under the newly named “Conclusions” section.

I recommend keeping the conclusion as a separate chapter and writing this more concisely.

Response: As mentioned above, we have now kept conclusions as such and removed the limitations portion. If you did not mean that doing this would keep the conclusion more concise, please let us know and we will make further adjustments to either section.

Reviewer 2 Report

The present study compares different blood flow restriction training devices. The manuscript is well written according to English grammar and languages. 

The introduction offers a wide view of the topic, naming the kaatsu original devices and the existing research about the topic.

In Methods, I suggest the authors to include a "Design of the study" section explaining the type of study, the participants recruitment and selection. Please consider to follow a  reporting data guide according with the type of study. 

Line numbers hide names in table 2.

Please fill the Institutional Review Board Statement section at the end of the manuscript. It seems as if there is not an ethics approval for this study.

Was this trial previously registered? Please indicate register and code.

Apart from these importante issues, the manuscript could be interesting for readers. Author must clarify those point before processing further.

Author Response

Dear Reviewer #2,

We massively appreciate both your positive and constructive feedback and have made our utmost efforts to make corresponding edits when able. We wholly believe that your suggestions have enhanced the present manuscript draft and thus we look forward to any future advice you might have our new iteration.

The present study compares different blood flow restriction training devices. The manuscript is well written according to English grammar and languages. 

The introduction offers a wide view of the topic, naming the kaatsu original devices and the existing research about the topic.

In Methods, I suggest the authors to include a "Design of the study" section explaining the type of study, the participants recruitment and selection. Please consider to follow a  reporting data guide according with the type of study. 

Response: An “2.1. Experimental Approach to the Problem” section was added and now precedes the “Participants” section. Predicated on the format of previous publications our group have in fellow MDPI-associated journals (please see links below in the response to your comment on clinical trial registration), we contend to retain participant recruitment and selection within the “2.2. Participants”. Please do not hesitate to let us know if any additional information is warranted in this section that is not adequately described in the remaining methods sub-sections.

Line numbers hide names in table 2.

Response: The Oxygen editors were contacted to resolve this formatting issue and thus you should find that no line numbers obscure table text in table 2, nor any other table.

Please fill the Institutional Review Board Statement section at the end of the manuscript. It seems as if there is not an ethics approval for this study.

Response: The IRB statement section is now filled out in its respective area and this information is now reiterated in the methods section as well.

Was this trial previously registered? Please indicate register and code.

Response: The present study was part of a larger series of investigations that were aimed at assessing performance versus clinical outcomes and thus we asserted that a clinical trial registration was unwarranted. These studies are now published in the MDPI journals, Nutrients (PMID: 36501070, PMID: 35276860) and Muscles (https://doi.org/10.3390/muscles1030013). In brief, we contended our observations on exercise-associated parameters does not directly focus on health-related outcomes as the NIH definition describes. We contend the same narrative for the present manuscript.

We have also have extensive discussion on this topic with other editors for additional high-impact exercise science journals and can provide their sources, but their description essentially argues that the only time that we would need to provide a trial registry number is if the investigation was government funded or if the primary outcomes is a disease or condition modification. If you need additional clarification or have further concerns regarding this issue, please let us know and we would be happy to discuss further.

Apart from these importante issues, the manuscript could be interesting for readers. Author must clarify those point before processing further.

Round 2

Reviewer 2 Report

Thank you for all the responses and interest.